# Factors Influencing Green Innovation Adoption and Its Impact on the Sustainability Performance of Small- and Medium-Sized Enterprises in Saudi Arabia

Mohammad Wasiq [1], Mustafa Kamal [2] and Nazim Ali [3,*]

1 College of Administrative and Financial Sciences, Saudi Electronic University, Riyadh 11673, Saudi Arabia
2 Department of Basic Sciences, College of Science and Theoretical Studies, Saudi Electronic University, Dammam 32256, Saudi Arabia
3 Department of Geography, Zakia Afaque Islamia College, Siwan 841226, India
* Correspondence: nazimamu.geography@gmail.com

**Abstract:** Due to the growing worries of communities and governments about the depletion of natural resources and environmental contamination, green innovation (GI) is currently receiving much attention on a global scale. This study intends to investigate how Green Innovation Adoption (GIA) influences Sustainable Performance (SNP) in Saudi Arabia's small- and medium-sized enterprises (SMEs) based on primary research. A conceptual framework model is developed to better comprehend the relationships of Government Support (GS), External Partnership and Cooperation (EPC), Rules and Regulatory Factors (RR), Market and Customer Factors (MC), Organization and Human Factors (OH), Green Innovation Strategy (GIS), and Technology Factors (TF) with GIA. The evaluation of hypotheses is performed using the Partial Least-Squares Structural Equation Modeling (PLS-SEM) method. The study's findings are obtained using the SPSS 24.0 and AMOS 24.0 software programs. The results of this study reveal that GS, EPC, RR, MC, OH, and TF all have a positive impact on GIA. Furthermore, it has been noted that GIA has a positive impact on the economic, social, and environmental performance of SMEs in the Kingdom of Saudi Arabia. In accordance with the findings, corporate units that use GI would produce more acceptable eco-friendly and long-term performance.

**Keywords:** green innovation adoption; sustainable performance; PLS-SEM; Kingdom of Saudi Arabia



## 1. Introduction

The degradation of the environment is a serious threat to the natural ecosystem and to the economic growth of humans [1]. It affects both the sustainable economic development (SED) and performance of manufacturing enterprises [2,3]. To protect their natural environments, governments and business firms are switching towards integrating sustainable processes and sustainable manufacturing methods into their business [4]. The ever-increasing environmental issues and tasks have colored the role of SMEs. SMEs are labelled as the backbone of economies because they act as the drivers of innovations to small and local economies, create employment opportunities, and help in manufacturing of products [5,6]. On the very opposite side, more than two-thirds of industrial pollution is contributed by SMEs due to the fact that these industrial units attach lesser importance to protection of the natural environment [7]. In developing countries of the world, SMEs have a tendency to use Conventional Production Methods (CPM) and give less attention to environmental preservation and protection activities. CPM are perilous to the health of the environment [8–13].

As per Sustainable Development Goal No.13 in the 1st Voluntary National Review "Towards Saudi Arabia's Sustainable Tomorrow" of the KSA submitted to the UN High-Level Political Forum on 9–18 July 2018 in New York, climate change is a great challenge globally in the twenty-first century, with far-reaching long-term effects on the Earth's

ecosystems [14]. The KSA is working to improve living circumstances, with the climate playing a particularly important role. The KSA has promoted the use of GI for its goals, such as the utilization of renewable energy and the building of green structures. In the field of business operations, GI is the most significant strategy for environment protection, involving reduction in resource consumption, prevention of environmental pollution, production procedures, and adoption of management systems for the environment. According to [15], GI has been observed in prevention of environmental pollution, industrial waste reduction, and the adoption of systems focused on environmental management. Additionally, GIs are seen as significant elements of social, economic, and environmental growth.

Business operations that pose a threat to the stability of the environment need to implement eco-friendly approaches for sustainable development, and one such approach is the adoption of GI [15–19]. GI can be defined as the steps and measures taken to lessen the damaging and destructive effects that business operations and production processes may have on environmental quality by giving greater priority to the advancement of products, processes, technologies, and management methods [13,20,21]. GI is used interchangeably with environmental innovation and eco-innovation [22–24]. It focuses on development of environmentally friendly products and processes through the adoption of organizational practices including use of green raw materials, eco-design principles, lesser utilization of raw materials, and minimization of emissions [20,25–29]. Moreover, green products are eco-friendly goods, while green processes include innovative methods, tools, and techniques that also produce environmentally friendly products [3,30]. Furthermore, GI is the most significant strategy for environment protection, involving reduction in resource consumption, prevention of environmental pollution, production procedures, and adoption of environment management systems in the field of business [15,31].

A large number of commercial firms are under inescapable pressure to use green innovation techniques in order to achieve SED [20,21]. Market innovation improves the financial and economic performance of business firms, particularly SMEs, and thereby increases business image and profitability [32]. GI in products reduces the financial risk and increases the performance and effectiveness of enterprises [11,33–36]. According to [37], SMEs largely depend on commercial benefits and GI motivation practices. The inclination and disposition of SMEs to espouse GI will result in growth acceleration, environmental protection, and social satisfaction [27,37]. Enterprises' corporate GI planning and strategy have a positive and significant impact on the uptake of green innovation [13,28,38–40]. Generally, innovation implies adoption of new procedures and practices by business firms, including products, materials equipment, services, and policies, with the purpose of increasing the productivity and effectiveness of business firms [7,33].

A different technique may be more suitable to assess SMEs [41,42] given their distinctive and varied characteristics, even if stakeholder theory [43] has been frequently utilised to study large corporations. However, as numerous SME studies have noted, SMEs' corporate social responsibility (CSR) initiatives have received very little attention from researchers and other interested parties [41,44,45]. Four explanations were put out by [45] as to why researchers have not paid as much attention to SMEs: SMEs lack access to CSR activities, they have fewer resources accessible to them, there is a paucity of SMEs with the visibility to get the attention of academics, and there is no research methodology that is suitable to SMEs [46].

A full GI measurement is required for a number of reasons. It aids policymakers in understanding the overall pattern of GI activity, identifying key drivers and barriers, and establishing suitable policies and framework conditions as a consequence. It may also serve to encourage environmental consciousness in enterprises, particularly if the data highlights the benefits to firms and industries. Concerns about the environment have an influence not just on economic growth but also on company performance. As a result, it is vital to study the most important aspects influencing SMEs in the KSA. The study attempts to analyse the factors influencing the adoption of GI and its influence on the sustainability performance of SMEs in Saudi Arabia.

Plenty of research has been devoted to the causal relationship between GIA and its impact on the SNP. However, not much attention has been paid to investigating the associations of GS, EPC, RR, MC, OH, GIS, and TF with GIA. Further, the impact of GIA has been witnessed on economic performances (EP), environmental performances (ENP), and social performances (SP) in Saudi Arabia. Adoption of GI is a widely accepted and applied phenomena to reduce pollution, which the present study intends to test.

The study is structured as follows: Section 2 focuses on hypothesis formulation. Section 3 includes data collection methods, research instruments, sampling procedures, and statistical approaches. Section 4 deals with results, data analysis, and hypothesis testing, while Section 5 contains conclusions and policy implications. Finally, Section 6 discusses limitations of research and future research. The flow chart in Figure 1 depicts the overall research framework for this study.

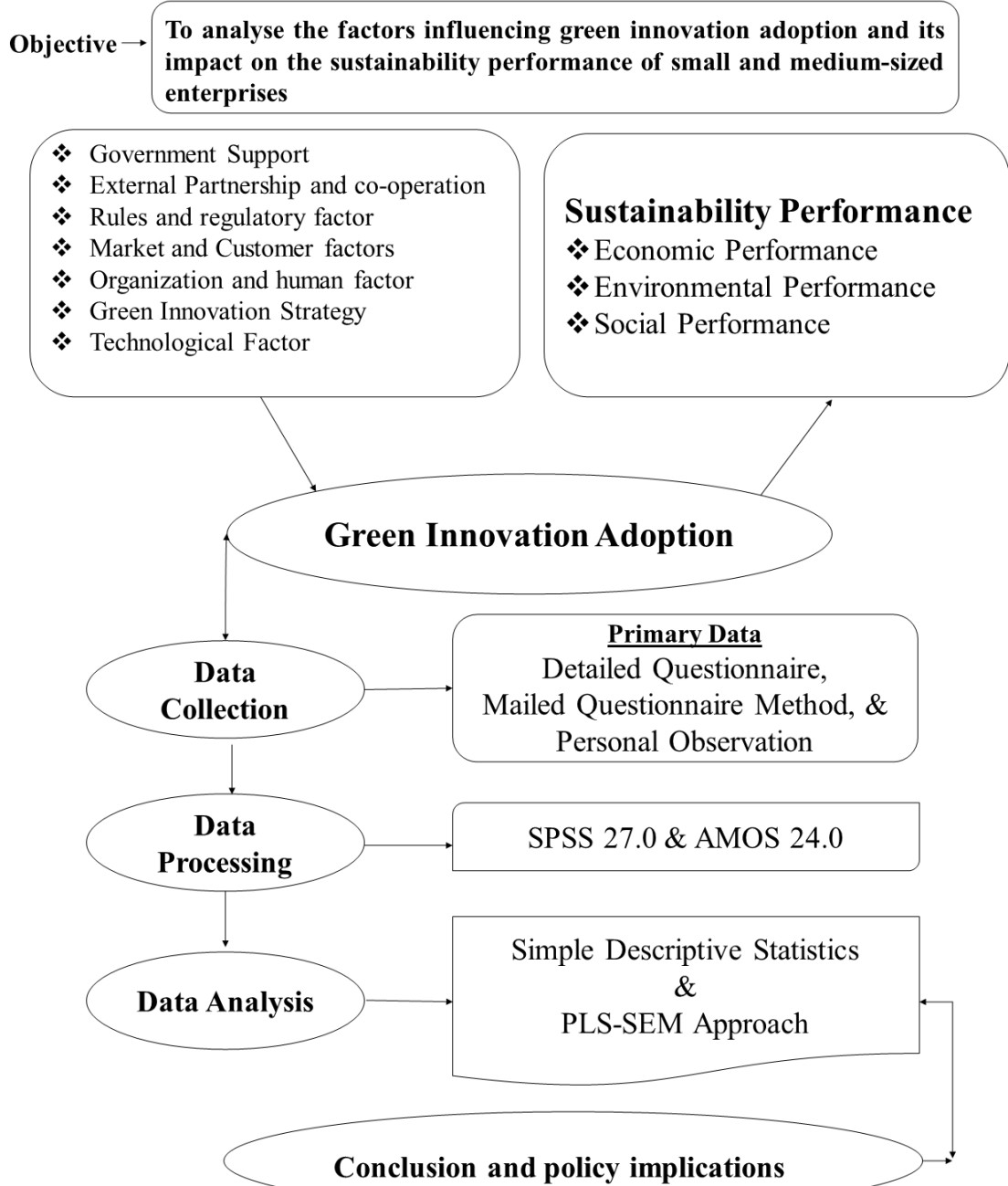

**Figure 1.** The flow chart of the research framework.

## 2. Literature Review

Several previous studies indicate that enterprises with green innovation (GI) practices are more popular than their direct competitors in overall performance, as they maximize the use of their available green resources and their ability to quickly respond to the needs of consumers and contribute substantial value and resources to the organization. In [8,47–49], GI is defined as a contemporary and noble method for managing processes and manufacturing that may reduce environmental concerns and pollution while minimizing negative effects on resources and energy usage. There is clear association between the adoption of green standards and the commitment of top management, as internal factors such as management commitment, relationships with suppliers, and regulatory and consumer pressure influence green purchasing adoption. GI can be implemented via the adoption of GI practices such as the use of green and fewer materials during product design and the reduction in water, energy, and other raw materials consumption. The authors of [2] used the PLS-SEM method and found that GI strategy is positively influenced by GI adoption in SMEs in Indonesia, and they proposed that firms should develop a GI strategy in order to gain environmental organizational legitimacy and better GI performance. The authors of [50] explore the relationship between absorption capacity and GI adoption in SMEs using the PLS-SEM approach, as well as the mediating influences of sustainable capabilities such as sustainable orientation, sustainable human capital, and sustainable collaboration. The authors of [51] analyzed data from Ghanaian manufacturing enterprises to explain the relationship between green supply chain integration, manufacturing practices, and SNP using the PLS-SEM model. Furthermore, [25] showed that sustainability variables, such as environmental and economic performance, have a substantial positive effect on GI processes based on the Malaysian hotel sector using the PLS-SEM model. According to [46], green transformational leadership has a limited effect on GI and has an indirect influence on the industry's environmental performance through GI. Using the PLS-SEM model, [21] investigated the factors related to GI adoption in SMEs in Pakistan. Based on a study of SMEs in Thailand, [52] argues that GI has a positive and significant effect on the competitive market benefit of green product lines and the progress of innovative green products, whereas [5] concluded that GI adoption has the strongest effect on SNP, including economic, social, and environmental performance.

## 3. Hypothesis Development

### 3.1. Impact of GS on the GIA

The policies formulated by the government, mostly in developing countries of the world, play a crucial role in encouraging SMEs to adopt environmentally friendly production methods. With the involvement of government in policy making in South Korea and Malaysia, the SMEs shifted their production processes from traditional to modern technologies [53]. Therefore, government intervention policies are currently needed to persuade SMEs to adopt GI techniques and technologies [54,55]. In the context of Cambodia, it was observed that government plays a moderating role between internal factors (market strategies, entrepreneurial values, and management) and growth performance of SMEs. However, there is not even a single study in the context of SMEs in Saudi Arabia that examines the moderating role of government intervention between TF and GI adoption [56]. Financial incentives, training programs, pilot projects, and technical resources are encouraging and stimulant factors for SMEs to switch towards green [34,49,57–59]. Keeping these factors in consideration, the study has proposed the following hypothesis:

**H1:** *GS influences Saudi Arabia's GIA in a positive way.*

### 3.2. Impact of EPC on the GIA

Promoting green practices in SMEs requires EPC, which is a crucial aspect. The capabilities of firms are regarded as key market resources, and inter-firm collaboration has grown in importance as a method of organizing and using marketing resources for

enhanced competitive advantage and internal and external knowledge sharing [16,17]. Without harming the environment, businesses need the cooperation and interdependence of clients, distributors, suppliers, and universities [60]. Therefore, the following hypothesis has been formulated:

**H2:** *EPC influences Saudi Arabia's GIA in a positive way.*

### 3.3. Impact of RR on the GIA

Regulatory burden on small- and medium-sized enterprises is a key factor in ensuring the adoption of GI by SMEs [22]. Similar to this, [61] found a substantial correlation between RR and environmental sustainability, as well as a positive correlation with the adoption of GI by businesses. Additionally, the pressure from environmental regulations compels firms to embrace GI, improving their cost-competitiveness and productivity/profitability. Furthermore, [22] observed in their work that RR increases the proportion of adoption of GI in SMEs concerning environmental safety and its proper implementation. Therefore, the following hypothesis has been formulated:

**H3:** *RR influences Saudi Arabia's GIA in a positive way.*

### 3.4. Impact of MC on the GIA

People, tools, technology, management skills, culture, and processes are all needed in the effort to connect with and manage sustainability holistically. Every one of these factors incorporates a sustainable culture and strategy that might improve business performance [62,63]. Further, culture and leadership are crucial resources for ensuring SNP with these resources. Green products can work as an inducement for companies to capture a significant market share. According to [64], one way to encourage SMEs to invest money in GI is through the desire of the customers to pay more attention to green innovative products. Customers are also the products' final users. Therefore, their demands may have a greater impact than other factors in encouraging producers to embrace GI. In medium-sized enterprises, the link between GI and environmental performance is higher than that of small businesses [31]. Presently, the number of green consumers is increasing and the awareness regarding green product development is also increasing [10]. Therefore, the following hypothesis has been formulated:

**H4:** *MC influences Saudi Arabia's GIA in a positive way.*

### 3.5. Impact of OH on the GIA

''Meeting the demands of a firm's direct and indirect stakeholders (such as shareholders, employees, customer pressure groups, communities, etc.) without jeopardising its potential to satisfy future stakeholder needs as well" is how organisational sustainability is defined [65]. Various studies are of the opinion that organisations who rely on GI are highly prosperous and thriving in comparison to their competitors because they leverage their green resource pool to promptly respond to the need of customers [48]. Several internal factors including customer and regulatory pressures, partnership with suppliers, and management commitments influence a firm's acquisition of green purchasing. Hence, there exists a direct relationship between adoption of green technology and management commitment [66]. Green technology adoption and management attitude are significantly correlated. In addition, GIA and overall ENP were recorded as positively correlated with each other [67].

On the other hand, human resource management (HRM) positively affects products and GI, and the practices of HRM with a focus on encouraging commitment culture have positive and significant effects on the innovative orientation of firms [21,56]. Moreover, [68] finds that strategic human resource management positively impacted product innovation in firms that have progressive culture and flat structure. In the same way, [69] noted that

in comparison to technological and product innovation, HRM does not have a strong effect on managerial and innovation processes. Hence, past studies shows that there exist mixed results on the association between HRM and GI in firms. Therefore, the following hypothesis has been formulated:

**H5:** *OH influences Saudi Arabia's GIA in a positive way.*

### 3.6. Impact of GIS on the GIA

To bolster the GI, environmental strategy acts as a means of propulsion, with due focus on internal and external environments [70]. With proactive GI procedures, business firms produce environmentally friendly innovations [71]. GI strategies can help firms to minimise the socio-economic costs of environmental damage on one hand, but on the other hand, it helps them to take advantage of market opportunities and competition [40]. Eventually, with such a proactive environmental plan, the environmental performance of such firms will improve [64]. Application of such strategies will encourage environmentally friendly concepts in the designing and packing process, thereby promoting GI. From the above findings, the following hypothesis has been formulated:

**H6:** *GIS influences Saudi Arabia's GIA in a positive way.*

### 3.7. Impact of TF on the GIA

The fast development in the field of science and technology has revolutionized the world and brought improved convenience to human life, but on the gloomy side, it has caused destruction of ecological environment, climate change, global warming, and exploitation of resources [72,73]. To overcome this challenge, the need for GI comes into the picture, which has now become a global discussion [4]. This contemporary philosophy involves the usage of advanced technologies to reduce pollution levels and protect the natural environment as well as improve resource utilization [74–77]. Although technological developments have modernized our lives on another side, they have hurled us into the depths of catastrophe in the form of resource scarcity, environmental degradation, and climate change. In spite of negative effects of technology on the quality of the environment, simultaneously, it provides a solution to address environment-related issues including climate change, food security, waste management, and numerous global challenges [78,79]. Environmental technologies include new, cleaner, and green technologies that aim at minimization of negative impacts of technology and resource use. In addition to efficient utilization of resources, technological innovations play a significant and positive role in the growth and development of renewable energy sources [80–83]. To overcome the menace of environmental degradation, eco-innovations are better solutions and important instruments. They provide cleaner technologies and better business management models, which are based on scientific knowledge and effective monitoring [84].

SMEs are much more adaptive to market fluctuations and distortions and highly resistant to technological changes as compared to giant enterprises [85]. In response to the pressures of stakeholders, SMEs have very lately adopted GI initiatives [50]. Only a negligible amount of literature has focused on the role of technological factors in the adoption of GI and concluded that TF have significant effect on adoption of GI [55]. Friability, compatibility, complexity, and relative advantage are the important technological characteristics that significantly affect innovation adoption [86].

Until now, research on the role of TF in GIA for Saudi Arabian SMEs has been incomplete and almost missing, and this study is an effort to fill this research gap. Furthermore, the study analyses the intermediating role of government intervention (an external factor) in GIA. The identification of TF in GIA is important because policy makers and SMEs must have familiarity with such factors. Then and only then, they can find plausible solutions and step smoothly towards a green phase. With changing technology, there is a need to reduce the life cycle of products and uphold economic competitiveness, which is possible

only when business firms invest in environmental innovations [43]. Green technology adoption improves competitiveness and performance of firms [66,87,88]. Lastly, green technology adoption improves the business reputation of a firm by allowing it to gain environmental certification. Therefore, the following hypothesis has been formulated:

**H7:** *TF influences Saudi Arabia's GIA in a positive way.*

Sustainability involves the convergence of three important aspects, namely environment, economy, and society, in the Triple Bottom Line Model (TBLM) [71]. Moreover, environmental quality, economic profitability, and social welfare are three important measurements of human well-being [89]. However, the industrial sector usually focuses on only the economic aspect and altogether denies the other two aspects [90]. Undoubtedly, sticking only to the economic aspect and altogether neglecting the environmental and social aspect is not sufficient for achieving sustainability [91,92].

### 3.8. Impact of GIA on the EP

The EP involves meliorating marketing and financial capabilities with the adoption of green techniques and strategies, which improve the economic position of business organisations above the industrial average, so that they have their own economic system across which they can conduct their business [88,93–96]. Several business firms have switched to cleaner technologies such as renewable energy sources to replace non-renewable energy sources for sustainable economic growth [97]. Relying on GI will definitely reduce organisational costs, including energy consumption costs and waste management costs [61]. Finally, with GI practices, the comparative as well as competitive advantages of business organisations will be improved, which in turn will result in better performance [98,99]. Therefore, the following hypothesis has been formulated:

**H8:** *GIA influences Saudi Arabia's EP in a positive way.*

### 3.9. Impact of GIA on the SP

The business companies and industrial units that invest in social accountability can reap better fruits through recruitment of good and efficient staff, innovations, and consumer satisfaction, which in turn play a pivotal role in their SP [100]. Awareness among the staff regarding social responsibilities, suitable communication with staff, staff retention, and satisfied staff assist business companies in dealing with ever-increasing environmental issues [101].

**H9:** *GIA influences Saudi Arabia's SP in a positive way.*

### 3.10. Impact of GIA on the ENP

The ENP of a business firm is defined as the minimisation of environmental incidents and curtailment of poisonous and dangerous substances [61]. Several industrial organisations are giving deeper attention to strategic environmental performance to actualize competitive benefits [60,102]. Environment legislation and market pressures have improved awareness and apprehensions among business organisations with regard to environmental performance. The implementation of environment-associated programs will alert a significant number of industrial units and business firms to reduce solid waste, greenhouse gas emissions (GHGs), and other dangerous chemicals [103]. The above-mentioned findings justify the viewpoint that improvement in production processes in turn results in better environmental performance [104].

**H10:** *GIA influences Saudi Arabia's ENP in a positive way.*

Figure 2 depicts the detailed research model that explains the relationship between all of the constructs in the current study. And how the various hypotheses H1, H2, H3, H4, H5, H6, H7, H8, H9, and H10 are related to one another.

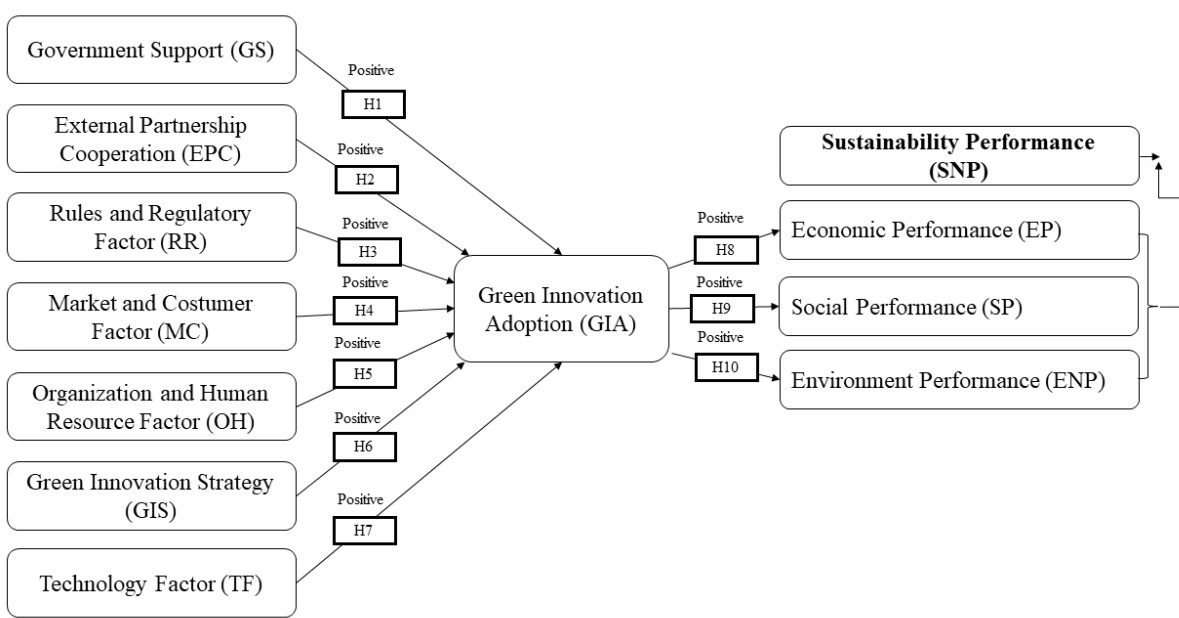

**Figure 2.** Relationship among all constructs and the study's various hypotheses.

Table 1 presents socio-demographic characteristics of sampled households. Out of the total sample households (N = 394), a majority (55.33%) were males, while the remaining respondents were females. The age classification of the households ranges from 40 to 50 years of age (31.22%) followed by above 50 years (15.89%) of age, while 23.71% were in the below-30-years-of-age group. Further, regarding education level of respondents, 44.162% of respondents were graduate, and 19.543% were post-graduate, while only 2.538% respondents were educated up to middle school (see Figure 3). Among the respondents, 26.63% were CEOs, followed by directors (25.86%), managers (15.77%), executives (16.23%), and supervisors (12.69%) of enterprises, while 2.82% were found to be others. The category of others includes third- and fourth-grade employees of enterprises.

**Table 1.** Socio-demographic characteristics of respondents.

| Variable | Percentage |
|:---:|:---:|
| Gender | |
| Male | 55.33 |
| Female | 44.67 |
| Age | |
| Less than 30 | 23.71 |
| 30–40 | 29.18 |
| 40–50 | 31.22 |
| Above 50 | 15.89 |
| Positions | |
| CEO | 26.63 |
| Director | 25.86 |
| Manager | 15.77 |
| Executive | 16.23 |
| Supervisor | 12.69 |
| Other | 2.82 |

Source: Based on primary data.

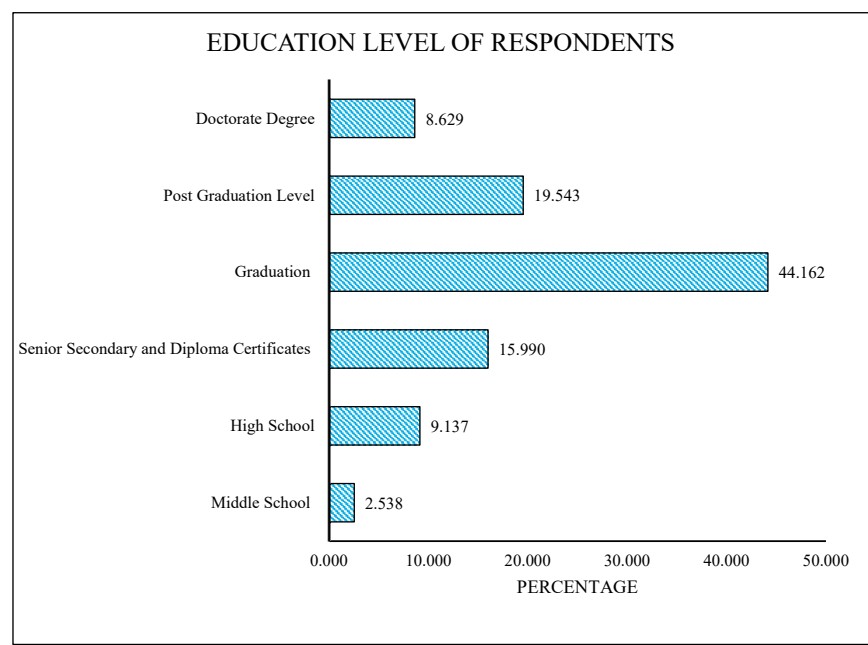

**Figure 3.** The respondents' educational background in Saudi Arabian SMEs.

## 4. Enterprise Profile

Concerning the ownership structure of enterprises, as seen in Table 2, about 51.79% of respondents were private/own enterprises, followed by joint venture enterprises (36.67%) and state-owned or state holding enterprises (11.54%). Further, 32.47% of enterprises had been operating for less than 5 years, followed by those that had been operating for 5 to 10 years, at 30.98%. About 35% of enterprises belonged to the petroleum industry, followed by textile, food and beverage, automobile, chemical, oil and gas, paper and pulp, plastic, building material and construction, biotechnology and medical care, electrical, and transportation industrials (see Figure 4). About 25.15% of enterprises have 31 to 50 employees, while 0.77% of enterprises have more than 300 employees.

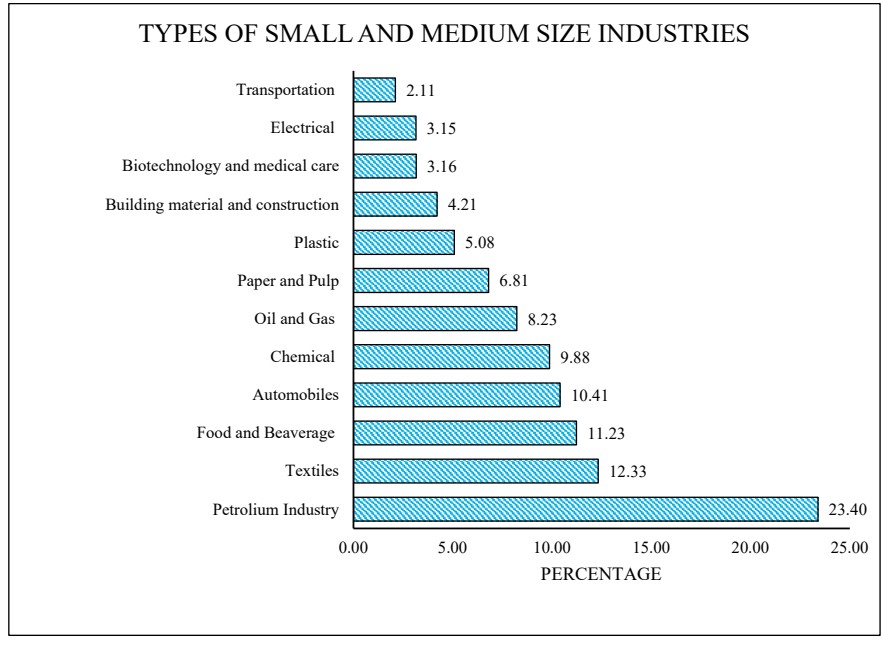

**Figure 4.** Different Kinds of SMEs in Saudi Arabia.

**Table 2.** Enterprise Profile.

| Ownership Structure of Enterprise | Percentage |
|---|---|
| State-owned or state holding enterprise | 11.54 |
| Private/own enterprise | 51.79 |
| Joint venture | 36.67 |
| Number of years operating | - |
| Less than 5 | 32.47 |
| 5–10 | 30.98 |
| 11–15 | 16.51 |
| 16–20 | 14.69 |
| Above 20 | 5.34 |
| Number of employees | - |
| 1–5 | 6.33 |
| 6–10 | 9.11 |
| 11–30 | 24.32 |
| 31–50 | 25.15 |
| 51–100 | 22.14 |
| More than 100 but fewer than 300 | 12.17 |
| More than 300 | 0.77 |

Source: Based on primary data.

## 5. Methods of Data Collection, Instruments, Sampling Procedures, and Statistical Techniques

*Research Design*

A quantitative analysis based on a cross-sectional survey has been performed to investigate the factors influencing GIA and its impact on the sustainability performance of small- and medium-sized enterprises in the Kingdom of Saudi Arabia.

## 6. Data Collection

The present study is based on both primary and secondary sources of data. This study relies on primary data sources, and the data are collected through a well-structured questionnaire designed based on the previous literature. The five-point Likert scale (including (1) strongly disagree, (2) disagree, (3) neutral, (4) agree, and (5) strongly agree) is used to design and develop a questionnaire. Further, we sent out 750 questionnaires to different small- and medium-sized enterprises (SMEs) online via e-mails and Google-form URLs. Out of the total sent questionnaires, 394 were recovered for the analysis. These questionnaires were sent to CEOs, managers, directors, and departments in charge of environmental protection related to production and operation due to the fact that these persons have first-hand and in-depth information and knowledge regarding green practices, the present situation of the enterprises, and SNP of industries. For the data collection, we used snowball random sampling techniques. It is the most suitable approach for collecting first-hand information because every unit has an equal chance of selection.

*Research Instrument, Reliability, and Convergent Validity*

For data collection, a structured questionnaire was designed with the help of available literature. The questionnaire consists of statement-based questions focusing on critical factors determined in the use and adoption of GI and sustainability performance in SMEs in the KSA. To measure the responses, the five-point Likert Scale was used. To check the in-

ternal consistency reliability of data, Cronbach's alpha and composite reliability were used. The convergent validity of data was measured by the average variance extracted (AVE).

## 7. Data Analysis

The SPSS 24 was used to conduct descriptive statistical analysis and to explain the demographic characteristics of the sample. The PLS-SEM, a broad-range multivariate method, was employed via AMOS 24.0 to statistically examine complex multivariable relationships [105].

## 8. Results and Discussion

### 8.1. Data Analysis, Results, and Hypothesis Testing

The SPSS software edition 27.0 and AMOS 24.0 were used for data analysis. SPSS 27.0 was applied to investigate the respondent's characteristics, correlation (association among constructs), and descriptive statistics (mean and standard deviation). AMOS 24.0 is a SEM software that was employed to test the reliability and validity of the observed variables with the purpose of testing the hypothesis.

In order to ensure validity, items were adopted from available literature as follows: GIA [10,21,23,106]; GS [107,108]; EPC [21,46]; MC [21,29,38,109]; OH [3,21]; RR [31,55,68,74]; GIS [15,39,71]; TF [16,110]; EP [33,70,94,95,98]; SP [37,52,91,100,101]; ENP [29,104]. The details of constructs/items are provided in Table 3.

**Table 3.** Measures of constructs.

| Constructs | Items | Mean | SD | LF |
|---|---|---|---|---|
| Green Innovation Adoption (GIA) | GIA-1: Materials that consume less energy and emit the least amount of pollutants are used in the production process. | 3.542 | 0.934 | 0.792 |
| | GIA-2: The item is simple to disassemble, recycle, and reuse. | 3.522 | 0.992 | 0.701 |
| | GIA-3: The manufacturing process lowers hazardous substances and recycles trash. | 3.389 | 0.981 | 0.707 |
| | GIA-4: The least quantity of materials are used. | 3.552 | 0.970 | 0.737 |
| Government Support (GS) | GS-1: The government assists the practice with financial/technical assistance and logistical skills. | 3.517 | 0.997 | 0.751 |
| | GS-2: Government authorities provide assistance to the organisation for GI information. | 3.744 | 0.852 | 0.763 |
| | GS-3: For SMEs, credit approval for green practises is simple. | 3.759 | 0.854 | 0.781 |
| External Partnership and Cooperation (EPC) | EPC-1: Suppliers promote the development of environmentally friendly products. | 3.621 | 0.985 | 0.753 |
| | EPC-2: The business cooperates with other businesses to share knowledge about GI, and it has environmental partnerships or certifications with suppliers. | 3.591 | 0.799 | 0.797 |
| | EPC-3: Universities and other research institutions provide our organisation with information on environmentally friendly procedures. | 3.626 | 0.837 | 0.719 |

**Table 3.** *Cont.*

| Constructs | Items | Mean | SD | LF |
|---|---|---|---|---|
| Rules and Regulatory Factors (RR) | RR-1: For logistical operations, the government establishes environmental laws, and industrial associations demand that we abide by those regulations. | 3.532 | 0.858 | 0.746 |
| | RR-2: High demands are made of the company by worldwide environmental rules. | 3.645 | 0.935 | 0.764 |
| | RR-3: There are numerous calls for regional and municipal environmental restrictions for the business. | 3.773 | 0.776 | 0.782 |
| Market and Customer Factors (MC) | MC-1: For our clients, protecting the environment is a top priority. | 3.704 | 0.759 | 0.631 |
| | MC-2: An incentive for a company of gaining a sizable market share can be seen in green products. | 3.724 | 0.759 | 0.667 |
| | MC-3: Publicity and advertising for green products have greater potential for success. | 3.611 | 0.804 | 0.664 |
| Organization and Human Resource Factors (OH) | OH-1: Company is committed to increasing green behavior within the firm. | 3.613 | 0.763 | 0.667 |
| | OH-2: Company offers rewards to employees for obtaining green knowledge and behavior. | 3.621 | 0.731 | 0.689 |
| | OH-3: Employees can quickly pick up new technology and exchange expertise with one another. | 3.622 | 0.732 | 0.811 |
| Green Innovation Strategy (GIS) | GIS-1: ISO 14000 | 3.524 | 0.761 | 0.669 |
| | GIS-2: Capital and technology investment | 3.613 | 0.811 | 0.699 |
| | GIS-3: Waste destruction or containment, as well as modifications aimed at preventing contamination | 3.601 | 0.801 | 0.679 |
| Technological Factors (TF) | TF-1: Green technology potentially brings greater economic benefits with improved environmental performance. | 3.627 | 0.739 | 0.701 |
| | TF-2: Green technology potentially improves company credibility. | 3.672 | 0.782 | 0.841 |
| | TF-3: Green practices can be easily implemented into any organisational framework. | 3.661 | 0.764 | 0.839 |
| Economic Performance(EP) | EP-1: Costs of energy consumption are falling. | 3.609 | 0.809 | 0.699 |
| | EP-2: Garbage treatment has been made cost-effective with improved capacity utilization. | 3.631 | 0.787 | 0.697 |
| | EP-3: Penalty fines for environmental accidents are reduced. | 3.449 | 0.791 | 0.712 |
| Environmental Performance (ENP) | ENP-1: Company has received environmental certifications for increment in performance over the last five years. | 3.697 | 0.809 | 0.824 |
| | ENP-2: During the last three years, resource use, such as water, energy, and gas, has declined. | 3.617 | 0.907 | 0.757 |
| | ENP-3: Environmental compliance and standards are being improved and followed (i.e., emissions, waste disposal). | 3.521 | 0.905 | 0.754 |

**Table 3.** *Cont.*

| Constructs | Items | Mean | SD | LF |
|---|---|---|---|---|
| Social Performance (SCP) | SCP-1: In the last three years, client satisfaction and motivation have improved. | 3.849 | 0.847 | 0.789 |
| | SCP-2: More beneficiaries (disadvantaged persons) and environmental issues are addressed by our business. | 3.618 | 0.893 | 0.739 |
| | SCP-3: Our industry delivers social/environmentally responsible services. | 3.749 | 0.802 | 0.767 |

Source: constructed by authors. Note: SD: Standard Deviation, LF: Loading Factors.

### 8.2. Measurement Model Assessment

As a result, Table 4 demonstrates the extracted average variance, composite reliability (CR), and Cronbach's alpha (AVE). In this study, Cronbach's alpha was used to establish the internal consistency of each construct. To verify the accuracy of the data, the average variance extracted (AVE) is employed, and all of the results are over the threshold of 0.5 [111]. In addition, the discriminant validity (DV) is measured in this study using Fornell and Larcker's [112] criteria and the hetero-trait–mono-trait (HTMT) ratio (See Table 4). The square root of each construct's AVE is placed at the diagonal for evaluation in accordance with the criteria of [112], denoting that it had a higher correlation coefficient with other items than its highest correlation coefficient with other constructs, indicating discriminant validity. Similarly, the HTMT ratio is assessed using the HTMT criterion, which is 0.90 [113]. All of the constructs' HTMT values are less than 0.90, as shown in Table 4. As a result, discriminant validity results for each construct are satisfactory.

**Table 4.** Reliability and Validity Assessments.

| Constructs | Cronbach's Alpha | CR | AVE |
|---|---|---|---|
| Government Support (GS) | 0.818 | 0.917 | 0.603 |
| External Partnership and Cooperation (EPC) | 0.856 | 0.909 | 0.747 |
| Rules and Regulatory Factors (RR) | 0.843 | 0.901 | 0.728 |
| Market and Customer Factors (MC) | 0.846 | 0.883 | 0.712 |
| Organization and Human Resource Factors (OH) | 0.820 | 0.919 | 0.605 |
| Green Innovation Strategy (GIS) | 0.842 | 0.879 | 0.708 |
| Technology Factors (TF) | 0.786 | 0.852 | 0.663 |
| Economic Performance (EP) | 0.736 | 0.881 | 0.594 |
| Social Performance (SP) | 0.841 | 0.899 | 0.726 |
| Environmental Performance (ENP) | 0.855 | 0.906 | 0.746 |
| Green Innovation Adoption (GIA) | 0.913 | 0.937 | 0.786 |

Note: CR = composite reliability; AVE = average variance extracted.

### 8.3. Measurement Model

The uniformity or consistency of the measurement items or constructs is connected to reliability. Cronbach's alpha was employed in this study to determine the internal consistency of each construct. As shown in Table 4, the consistency or reliability of all constructions ranged from 0.852 to 0.919. All constructs are greater than the recommended threshold of 0.70, thus demonstrating internal consistency of all the research constructs. Validity is defined as the extent to which the measured items or constructs capture the variables. To ensure validity in the present study, both the convergent and divergent validities were measured. For convergent validity, the loadings of all the concerned items must be

greater than 0.6, and AVE and CR should be greater than 0.6 and 0.5, respectively [111]. As demonstrated in Table 4, AVE of all the constructs ranged from (0.597 to 0.786), and CR is above the recommended range (0.852 to 0.937). Therefore, the convergent validity is proved by the results of research constructs. Additionally, the square root of AVE for each construct was compared to the correlations of the neighboring constructs to assess the discriminant validity. Table 5 shows that the square roots of the AVEs in the diagonals were larger than the off-diagonal correlation, exhibiting further corroboration of discriminant validity.

**Table 5.** The discriminant validity criterion test by Fornell and Larcker.

|  | GS | EPC | RR | MC | OH | GIS | TF | GIA | EP | SP | ENP |
|---|---|---|---|---|---|---|---|---|---|---|---|
| GS | 0.863 |  |  |  |  |  |  |  |  |  |  |
| ECP | 0.629 ** | 0.767 |  |  |  |  |  |  |  |  |  |
| RR | 0.566 ** | 0.616 ** | 0.857 |  |  |  |  |  |  |  |  |
| MC | 0.416 ** | 0.517 ** | 0.467 ** | 0.811 |  |  |  |  |  |  |  |
| OH | 0.654 ** | 0.554 ** | 0.594 ** | 0.589 ** | 0.816 |  |  |  |  |  |  |
| GIS | 0.633 ** | 0.535 ** | 0.613 ** | 0.633 ** | 0.611 ** | 0.771 |  |  |  |  |  |
| TF | 0.613 ** | 0.633 ** | 0.595 ** | 0.613 ** | 0.589 ** | 0.673 ** | 0.853 |  |  |  |  |
| GIA | 0.626 ** | 0.526 ** | 0.617 ** | 0.526 ** | 0.602 ** | 0.681 ** | 0.526 ** | 0.763 |  |  |  |
| EP | 0.563 ** | 0.663 ** | 0.671 ** | 0.663 ** | 0.563 ** | 0.431 ** | 0.463 ** | 0.663 ** | 0.777 |  |  |
| SP | 0.413 ** | 0.513 ** | 0.423 ** | 0.431 ** | 0.435 ** | 0.513 ** | 0.613 ** | 0.513 ** | 0.682 ** | 0.813 |  |
| ENP | 0.651 ** | 0.689 ** | 0.621 ** | 0.611 ** | 0.643 ** | 0.659 ** | 0.631 ** | 0.539 ** | 0.599 ** | 0.586 ** | 0.852 |

Note: (1) GIA: Green Innovation Adoption; OH: Organization and Human Factors; EPC: External Partnership and Cooperation; TF: Technology Factors; RR: Rules and Regulatory Factors; GS: Government Support; GIS: Green Innovation Strategy; MC: Market and Customer Factors; ENP: Environmental Performance; SP: Social Performance; EP: Economic Performance; (2) ** $p < 0.01$ (two-tailed); (3) Off-diagonal correlations are smaller than the square roots of AVEs, which are denoted by boldface in parenthesis.

### 8.4. Structural Model Assessment

After the confirmation of validity of the results, the structural model was assessed. In this study, the $R^2$ value of GIA is 0.913, while the R2 adjected value of GIA is 0.911 (Table 6). Thus, the model effectively explains a large portion of the variation in GIA. To determine if the model was adequate for predicting the indexes of each latent construct, a cross-validated redundancy (Q2) was used [114]. To test additional model fit assessments (AMFA), this statistical method is helpful. In AMOS 24.0, the blinding process is used to calculate the projected importance of a variable. According to [105], the projective relevance of the model can be judged by whether the Q2 value is greater than 0. The current work's Q2 score of 0.221 indicates that the model has a high level of predictive significance (Table 7).

**Table 6.** R square ($R^2$) value of construct.

| Variable | $R^2$ | $R^2$ Adjusted |
|---|---|---|
| EP | 0.392 | 0.391 |
| ENP | 0.664 | 0.663 |
| SP | 0.651 | 0.650 |
| GIA | 0.913 | 0.911 |

**Notes**: EP—Economic Performance, ENP—Environmental Performance, SP—Social Performance, GIA—GIA.

**Table 7.** Cross-validated redundancy ($Q^2$) of construct.

| Variable | SSO | SSE | $Q^2$ (=1 − SSE/SSO) |
|---|---|---|---|
| GS | 1182 | 1182 | |
| EPC | 1182 | 1182 | |
| RR | 1182 | 1182 | |
| MC | 1182 | 1182 | |
| OH | 1182 | 1182 | |
| GIS | 1182 | 1182 | |
| TF | 1182 | 1182 | |
| EP | 788 | 614.119 | 0.221 |
| SP | 788 | 402.561 | 0.489 |
| ENP | 788 | 423.499 | 0.463 |
| GIA | 1576 | 467.533 | 0.703 |

*8.5. Hypotheses Testing*

In this study, SEM is used to test the theoretical correlations. Table 8 and Figure 5 present the structural parameters of the hypothesized relationships. In hypothesis H1, the result outputs reveal that GS positively and significantly predicted GIA with β = 0.795, CIU value = 0.682. In order to achieve sustainable development goals (SDGs), several governments have started special subsidies, schemes, and programs [31,54,109]. For instance, the Saudi Arabian government has committed a significant share of incentives for research and development in order to support green energy and actively promote GI, eco-innovation, eco-technology, environmental protection, and industrial structure optimization.

**Table 8.** Hypotheses testing/mediation test using a bootstrap with 95% confidence interval.

| PTH | β = Value | CIL | CIU | t-Statistics | *p*-Value | Result |
|---|---|---|---|---|---|---|
| H1: GS → GIA | 0.795 | 0.491 | 0.682 | 29.849 | 0.000 *** | Supported |
| H2: EPC → GIA | 0.815 | 0.277 | 0.878 | 41.543 | 0.000 *** | Supported |
| H3: RR → GIA | 0.731 | 0.435 | 0.748 | 21.567 | 0.000 *** | Supported |
| H4: MC → GIA | 0.788 | 0.559 | 0.734 | 23.853 | 0.000 *** | Supported |
| H5: OH → GIA | 0.811 | 0.291 | 0.782 | 34.372 | 0.000 *** | Supported |
| H6: GIS → GIA | 0.043 | -0.089 | 0.159 | 0.076 | 0.000 *** | Rejected |
| H7: TF → GIA | 0.781 | 0.467 | 0.821 | 22.831 | 0.000 *** | Supported |
| H8: GIA → EP | 0.745 | 0.590 | 0.795 | 20.667 | 0.000 *** | Supported |
| H9: GIA → SP | 0.831 | 0.521 | 0.895 | 49.997 | 0.000 *** | Supported |
| H10: GIA → ENP | 0.801 | 0.511 | 0.811 | 32.490 | 0.000 *** | Supported |

Note: (1) GIA: Green Innovation Adoption; GS: GS; EPC: External Partnership and Cooperation; RR: Rules and Regulatory Factors; MC: Market and Customer Factors; OH: Organization and Human Factors; GIS: Green Innovation Strategy; TF: Technology Factors; SP: Social Factor; SNP: Sustainability Performance; (2) Unstandardized coefficients reported; (3) *** = $p < 0.001$; (4) ns = not significant; Confidence Interval Lower (CIL); Confidence Interval Upper (CIU); Path Hypothesis Testing (PHT).

In H2, the link between EPC and GIA was positive and significant (β = 0.815). From the analysis, it is noted that EPC is considered a significant factor in enhancing the GI in enterprises. Moreover, suppliers' environmental alliances and offering of improvements for greening production and processes intensify green practices in businesses. The interdependence and cooperation between corporations, clients, distributors, suppliers, and

universities are critical if organizations wish to manufacture goods without hurting the environment [106].

**Figure 5.** Structural parameters of the hypothesized relationships.

In H3, RR has a positive and significant impact on GIA ($\beta$ = 0.731). Enterprises are compelled to employ GI as a result of regulatory pressure, which improves their cost efficiency and profitability. Moreover, strict rules and regulations, as well as their suitable application, improve the rate of GI uptake in SMEs. Several researchers identified a positive impact of RR on GIA [22,115].

In H4, MC was a significant and positive predictor of GIA ($\beta$ = 0.795). According to [116], green products can serve as a motivator for productions to gain a considerable market share. Green products can serve as a motivator for trades to gain a considerable market share [11]. The customers' desire to pay more for green products can be used to encourage manufacturers to invest more in GI [103,110]. Customer demand may have a greater impact on manufacturers' adoption of GI than other factors because customers are the products' final users, as demand is increasing turnover in an industry [21,67]. Despite the fact that more people are becoming environmentally conscious customers, more people still need to be made aware of green products.

In H5, OH was a significant and positive predictor of GIA ($\beta$ = 0.895). Human resources have been related with green management and operational management, and they must assess their contribution to organizational sustainability and improving the performance of companies [115]. Moreover, the employees may be encouraged to adopt green practices in the organization if they are rewarded for their efforts [117].

In H6, GIS has a positive and insignificant impact on GIA ($\beta$ = 0.043). This is because the role of environmental strategy in supporting GI is crucial, with a particular focus on the influence on the internal and external surroundings. Additionally, the organization of the many resources needed for green manufacturing operations can be facilitated by the implementation of a GIS, which will again result in the formation of GI [110].

In H7, TF was a significant and positive predictor of GIA ($\beta$ = 0.781). According to [21], when SMEs realize that green technology offers greater financial and economic benefits than existing technology, they are more willing to adopt GI [20,79]. In H8, GIA was a positive and significant forecaster of EP, with $\beta$ = 0.745. In H9, GIA was a positive and significant predictor of SP, with $\beta$ = 0.831. In H10, GIA was a positive and significant predictor of ENP, with $\beta$ = 0.801. Therefore, H1, H2, H3, H4, H5, H7, H8, H9, and H10 were supported, while H6 was rejected. According to several academics, incorporating GI into product

development and business processes can have a number of advantages, including improved corporate image, product differentiation, increased competitive advantage, return on investment, increased sales, and increased efficiency in resource use [17,28,118]. This finding also pertains to managerial groups in SMEs in the KSA, because it shows how implementing GI in a company's environment may benefit the business. This condition will enhance enterprises' ability to compete in the market while also enhancing their SP and ENP, which will further enhance the firms' goodwill and reputation [21,119].

## 9. Conclusions and Policy Implications

The adoption of GI is a growing issue on a global scale, and it has prompted companies to continue managing their green potential and implementing GIS with the intent of environmental preservation and protection, as well as increasing organizational performance. Therefore, the impacts of GS, EPC, RR, MC, OH, GIS, and TF are direct and positive on the GIA. Similarly, GIA also has a positive relation with EP, SP, and ENP. The data for this study came from a primary source and were acquired using a well-structured questionnaire. For the analysis of data, simple descriptive statistics and PLS-SEM approach were used. The findings show that GS, EPC, RR, MC, GIS, and TF all have a significant positive influence on GIA in SMEs in Saudi Arabia. Thus, the GIA also demonstrates the beneficial effects on economic, social, and environmental performance.

These findings suggest that business units that employ GI will achieve more acceptable eco-friendly and long-term performance. As a result, they will be able to meet industrial and governmental requirements while minimizing waste output and pollution. In Saudi Arabia, SMEs must grasp the need for environmental protection, which would definitely benefit them as well. GIA (GIA) also has a favorable impact on SMEs' sustainability performance (SNP). This situation will help to strengthen the market's competitive edge while also focusing on their EP, SP, and ENP, which will help to boost their companies' image and reputation. The results produced here are critical in understanding how GI might operate in SMEs.

## 10. Limitations and Future Scope of Research

When extrapolating the findings of this study to other settings, various limitations must be addressed. The hypotheses were tested and verified in the setting of Saudi Arabia, in a cross-sectional survey utilizing a questionnaire. This strategy limits the ability to infer causativeness in construct-to-construct interactions. As a result of the inability of the study to monitor dynamic changes in GI in the development process of SMEs, the conclusions are overstated. To ensure accurate results, a longitudinal study that investigates the connections over the course of a long period of time should be conducted. There are a number of variables of GI that are directly or indirectly linked to GIA of SMEs in Saudi Arabia. As a result, it is advised that future studies include the firms' other internal and external elements. The current study, on the other hand, examines the link between GIA and SMEs' performance and discovers that there is a positive correlation between adoption of GI and SMEs' SNP. More research is needed to look into other factors that influence the long-term performance of the same businesses. Furthermore, the sample employed in this study, as well as the evolution of a given technology, may change across industries and nations. Furthermore, this cross-sectional survey only covered manufacturing SMEs; however, research into services and non-manufacturing SMEs is still needed to have a deeper understanding of this essential sector.

**Author Contributions:** All authors equally contributed to the study. All authors have read and agreed to the published version of the manuscript.

**Funding:** The authors extend their appreciation to the Deanship of Scientific Research at Saudi Electronic University for funding this research work through the project number (8102).

**Institutional Review Board Statement:** Not applicable.

**Informed Consent Statement:** Not applicable.

**Data Availability Statement:** Data supporting the findings of this study are available upon request from the corresponding author.

**Acknowledgments:** The authors thank to the Deanship of Scientific Research at Saudi Electronic University for funding this research work through the project number (8102).

**Conflicts of Interest:** The authors declare no conflict of interest.

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
