# Peer review of "Factors Influencing Green Innovation Adoption and Its Impact on the Sustainability Performance of Small- and Medium-Sized Enterprises in Saudi Arabia"

_sustainability, doi:10.3390/su15032447_

Round 1

Reviewer 1 Report

Dear Authors, Please answer these questions:

1. In table 4, Cronbach Aalpha amounts are more than CR. It is not possible. Please check always CR > Cronbach alpha. Do you have any explanations?

Check these references:

Ebrahimi, P., Ahmadi, M., Gholampour, A. and Alipour, H. (2021), "CRM performance and development of media entrepreneurship in digital, social media and mobile commerce", International Journal of Emerging Markets, Vol. 16 No. 1, pp. 25-50. https://doi.org/10.1108/IJOEM-11-2018-0588

Hair, J.F.; Henseler, J.; Dijkstra, T.K.; Sarstedt, M. Common beliefs and reality about partial least squares: Comments on Rönkkö and Evermann. Organ. Res. Methods 2014, 17, 182–209. [Google Scholar] [CrossRef][Green Version]

2. Please put your original measurement models output and final SEM model outputs from AMOS software.

3. Would you please tell me why used Amos as an confirmatory software (to check relationship not impact).

4. Please put standard deviation for all hypotheses in table 8.

Best regards and merry christmas,

One of reviewers.

Reviewer 2 Report

Dear authors, 

Thank you. Very intriguing paper. Comments are required to be addressed.

1. Abstract should be improved and decreased - to be concise but informative

2. Please check citation rules in the text for MDPI

3. Pay attention to some grammar and format mistakes. Most of your hypotheses are missing the verb, or Capital letters are in the middle of the text, formatting (Italics, Bold), line 471 -  R2 adjected value
4. Please provide a questionnaire and measurement development details (how did you construct your questionnaire/resources)

5. Sustainable performance can be approached from many perspectives, and one is the economic aspect of it. Some studies that should be included to provide context further, and perspective to the topic

COVID-19 Pandemic Implications for Corporate Sustainability and Society: A Literature Review

Impact of entrepreneurial leadership and bricolage on job security and sustainable economic performance: an empirical study of Croatian companies during COVID-19 pandemic

Strategic Planning and Sustainable Innovation during the COVID-19 Pandemic: A Literature Review

6. Additionally, the energy sector is the one which is particularly interesting and could benefit from green innovation, but it is very volatile.  Please check if you can enrich your study in comparison with other countries

Impact of COVID-19 on stock price crash risk: Evidence from Chinese energy firms. Energy Economics, 101, 105431. 

Financial deregulation and operational risks of energy enterprise: The shock of liberalization of bank lending rate in China. Energy Economics, 93, 105047. 

7. More theoretical support is needed for some hypotheses. Consider Influence of Virtual CSR Co-Creation on the Purchase Intention of Green Products under the Heterogeneity of Experience Value

8. H6 was rejected-please provide more justification why it was rejected in your study and if it is in line with some previous studies. 

9. Please complete your H8, 9, 10  with theoretical support in your Hypotheses testing part

10. Otherwise, I suggest separating Hypotheses testing from the Discussion part to make statistical explanations and theoretical support of your results more clear.

Thank you

Reviewer 3 Report

English language of the paper MUST be improved. There are many grammatical errors which are simply unacceptable.

ALL Tables and Figures must be referenced within the text flow of the paper. Some are, but most are not.

Please reformulate the Conclusions. At the moment, they sound like the repetition of the Introduction.

Please see attached file for some basic proof-reading and contents improvement suggestions.

Reviewer 4 Report

Factors Influencing Green Innovation Adoption and its Impact on the Sustainability Performance of Small and Medium-Sized Enterprises in Saudi Arabia

Dear authors,

It is a pleasure to review your manuscript. Below, please find my comments and recommendations:

Abstract

- The abstract must be reviewed and improved. Please, include quantitative achievements from the findings of this study. Please limit your abstract to 250 words. 

- Please capitalize titles such as Green Innovation Adoption (GIA) and others. 

- What theory (theories) underpins this research?

Introduction

- The introduction should be revised. The authors have added an excellent background and related literature. Still, they should consider more details about the methodology, empirical results, and policy implications, especially if there are studies related to the research topic in Saudi Arabia. 

- The introduction should include the research problem, objective, and research’s novelty. 

- What theory (ies) support this paper?

- The aim of this research is not explained at the end of the introduction. 

- Why will this study be necessary for practitioners and scholars?

The literature review section could be more substantial; please revise it.

Hypothesis Development

- 3.1.: I need clarification on whether there are studies, policies, and regulations in Saudi Arabia that the authors can use to support H1 and all other hypotheses. If there is none, how the authors concluded H1 is positive?

- 3.2.: Each hypothesis should be supported by at least 150 words. Please review.

- 3.3.: “Stakeholder theory's foundation”; is this theory the most important one in your paper?

- Research model: Please, fix “costumer”. It must be “customer.”

- What is the theory (es) that support this study?

Data and methodology

- What scales did the authors use for this survey? 

- Can the authors include the questionnaire as an appendix?

- I have the following questions:

    • Based on what the authors selected participants? Is there a list? Based on a governmental agency? How?
    • If the questionnaire was sent to CEOs, managers, directors, etc., how the authors stratified the sample in terms of region, industry, etc.?
    • How was the sample structured? 
    • Did the authors conduct a pilot to ensure that participants understood the variables on the questionnaire?
    • What cities or regions were selected for this sample distribution? Can you explain the rationale?
    • What demographic questions were included in the questionnaire?

General comments on the methodology:

- This section should be improved. The whole project can be compromised if the methodology is not well structured, especially the results. 

- The authors MUST put in a lot of effort to respond to the questions and improve the understanding of this section.

Conclusion and future direction

- This part of the study must be significantly improved. 

Finally, I wish you the best in this peer-review process, 

Regards,

Round 2

Reviewer 1 Report

Dear authors, 

The paper is acceptable in current format.

Best regards,

One of reviewers

Author Response

Thanks for the valuable response

Reviewer 2 Report

Paper was improved. The figures of demographic info isare redundant. They can be added to the demographic table you already have.

Author Response

Comment: The figures of demographic info is/are redundant. They can be added to the demographic table you already have.

Response: Thank you for your insightful feedback. This is the better representation of the data. therefore, we put both the Table and Figure separately. 

Reviewer 4 Report

Dear authors, 

Thank you for the significant improvement to this paper. I consider the manuscript is now publishable on Sustainability.

Good luck with your review process.   

Author Response

Thank you for your knowledgeable input.